# Sketch and shift: a robust decoder for compressive clustering

**Ayoub Belhadji**                                                                    *ayoub.belhadji@gmail.com*
*Univ Lyon, ENS de Lyon, Inria, CNRS, UCBL, LIP UMR 5668, Lyon, France*

**Rémi Gribonval**                                                                    *remi.gribonval@inria.fr*
*Univ Lyon, ENS de Lyon, Inria, CNRS, UCBL, LIP UMR 5668, Lyon, France*

**Reviewed on OpenReview:** *https://openreview.net/forum?id=6rWuWbVmgz*

## Abstract

Compressive learning is an emerging approach to drastically reduce the memory footprint of large-scale learning, by first summarizing a large dataset into a low-dimensional sketch vector, and then decoding from this sketch the latent information needed for learning. In light of recent progress on information preservation guarantees for sketches based on random features, a major objective is to design easy-to-tune algorithms (called decoders) to robustly and efficiently extract this information. To address the underlying non-convex optimization problems, various heuristics have been proposed. In the case of *compressive clustering*, the standard heuristic is CL-OMPR, a variant of sliding Frank-Wolfe. Yet, CL-OMPR is hard to tune, and the examination of its robustness was overlooked. In this work, we undertake a scrutinized examination of CL-OMPR to circumvent its limitations. In particular, we show how this algorithm can fail to recover the clusters even in advantageous scenarios. To gain insight, we show how the deficiencies of this algorithm can be attributed to optimization difficulties related to the structure of a correlation function appearing at core steps of the algorithm. To address these limitations, we propose an alternative decoder offering substantial improvements over CL-OMPR. Its design is notably inspired from the mean shift algorithm, a classic approach to detect the local maxima of kernel density estimators. The proposed algorithm can extract clustering information from a sketch of the MNIST (resp. of the CIFAR10) dataset that is 10 times smaller than previously and much easier to tune.

## 1 Introduction

In an era where resources are becoming scarcer, reducing the footprint of learning algorithms is of paramount importance. In this context, sketching is a promising paradigm. This consists in conducting a learning task on a low dimensional vector called a *sketch* that captures the essential structure of the initial dataset (Cormode et al., 2011). In other words, the sketch is an informative summary of the initial dataset which may be used to reduce the memory footprint of a learning task.

This article studies decoding in the context of *compressive clustering*. This is an emerging topic of research in machine learning aiming to scale up the (unsupervised learning) task of clustering by conducting it on a sketch (Gribonval et al., 2021b). In this context, the sketch[1] is the mean of a given feature map over the dataset (Keriven et al., 2017), and *decoding* consists in the recovery of the cluster centroids from such a sketch. So far, this learning step was conducted using a heuristic decoder called CL-OMPR (Keriven et al., 2017). Despite existing proofs of concept for compressive learning with CL-OMPR, an in-depth examination of the properties of this decoder has been lacking.

The starting point of our study is a scrutinized examination of CL-OMPR. First, using a numerical experiment, we show how CL-OMPR might fail to recover the clusters even in an advantageous scenario where the dataset is generated through a Gaussian mixture model with well-separated clusters. Second, to provide

---

[1]The word "sketching" is highly overloaded in machine learning. In this paper it refers to a vector computed as in equation 1.

an explanation of this deficiency, we analyze the first step of the algorithm which consists in finding a new centroid that maximizes a correlation function. The latter is an approximation of the kernel density estimator (KDE) associated to a shift-invariant kernel when the feature map corresponds to a Fourier feature map. We show that the main shortcomings of CL-OMPR stem from optimization difficulties related to the correlation function. Third, we propose an alternative decoder which takes into account these difficulties and validate it on synthetic and real datasets. In particular, in the case of spectral features of MNIST, we demonstrate that our algorithm can extract clustering information from a sketch that is 10 times smaller.

This article is structured as follows. Section 2 recalls definitions and notions related to compressive clustering. In Section 3, we conduct a numerical simulation that highlights the limitations of CL-OMPR. In Section 4, we propose an alternative decoder that circumvents the shortcomings of CL-OMPR. Section 5 gathers numerical simulations that illustrate the improvement of the alternative decoder upon CL-OMPR.

## 2 Background

Clustering algorithms group elements into categories, also called clusters, based on their similarity. Numerous clustering algorithms have been proposed in the literature; see e.g. Jain (2010).

Beyond the classical Lloyd-Max approach (Lloyd, 1982), compressive clustering (Keriven et al., 2017) was showed to offer an alternative both well-matched to distributed implementations and streaming scenarios and able to preserve privacy, see e.g. Gribonval et al. (2021b). The approach aims at drastically summarizing a (large) training collection while retaining the information needed to cluster. This objective is similar to that of coresets (see e.g. Feldman & Langberg (2011); Guo et al. (2022)), but the approach is radically different. Coresets summarize a collection of training samples by selecting a limited subset of representative samples (or sometimes of *transformed* samples). In contrast, compressive clustering is somewhat more democratic: instead of selecting a few "meritorious" samples, it computes a unique vector, called a sketch (of dimension moderately *higher* than the dimension of each training sample), that depends *equally* on all samples. The computation of this sketch is designed to capture the relevant information of the whole training collection to perform a given clustering task.

For instance, in Keriven et al. (2018a), the authors compressed 1000 hours of speech data (50 gigabytes) into a sketch of a few kilobytes on a single laptop, using a random Fourier feature map. In this work, the sketch $z_{\mathcal{X}}$ of a dataset $\mathcal{X} = \{x_1, \ldots, x_N\} \subset \mathbb{R}^d$ was taken to be

$$z_{\mathcal{X}} := \frac{1}{N} \sum_{i=1}^{N} \Phi(x_i), \tag{1}$$

where $\Phi : \mathbb{R}^d \to \mathbb{C}^m$ is a given feature map. For instance, random Fourier features (RFF) are defined as $\Phi(x) = (\phi_{\omega_j}(x))_{j \in [m]} \in \mathbb{C}^m$, where

$$\phi_\omega(x) := \frac{1}{\sqrt{m}} e^{\mathbf{i}\langle x, \omega \rangle}, \tag{2}$$

and $\omega_1, \ldots, \omega_m$ are i.i.d. samples from $\mathcal{N}(0, \sigma^{-2}\mathbb{I}_d)$, with $\sigma > 0$ (Rahimi & Recht, 2007).

The sketch $z_{\mathcal{X}}$ can be seen as the mean of the generalized moment defined by the feature map $\Phi$, i.e., $z_{\mathcal{X}} = \mathbb{E}_{x \sim \pi_N} \Phi(x)$, where $\pi_N := \sum_{i=1}^{N} \delta_{x_i}/N$ is the empirical distribution of the dataset $\mathcal{X}$. In other words, $z_{\mathcal{X}} = \mathcal{A}\pi_N$, where $\mathcal{A}$ is the so-called sketching operator $\mathcal{A} : \mathcal{P}(\mathbb{R}^d) \to \mathbb{R}$ on $\mathcal{P}(\mathbb{R}^d)$, the set of probability distributions on $\mathbb{R}^d$, defined by

$$\mathcal{A}\pi = \int_{\mathbb{R}^d} \Phi(x) \mathrm{d}\pi(x). \tag{3}$$

A theoretical analysis of compressive clustering using random Fourier features was conducted in Gribonval et al. (2021a); Belhadji & Gribonval (2022). In particular, it was shown that the sketch size $m$ must depend on the ambient dimension $d$ of the dataset and the number $k$ of centroids in order to recover the centroids with high probability. More precisely, a sufficient sketch size was shown to scale as $\mathcal{O}(k^2 d)$, while numerical investigation in Keriven et al. (2017) showed that a sketch size $m = \mathcal{O}(kd)$ is enough in practice to recover the cluster centroids. Alternatively, data-dependent Nyström feature maps were shown to require a sketch

size that depends on the effective dimension of the dataset, which may be smaller than $\mathcal{O}(kd)$ (Chatalic et al., 2022). In these works, centroid recovery was achieved by addressing the following inverse problem

$$\min_{\substack{\boldsymbol{\alpha}\in\mathbb{R}_+^k;\sum_{i=1}^k \alpha_i=1 \\ c_1,\ldots,c_k\in\Theta}} \left\|z_\mathcal{X} - \sum_{i=1}^k \alpha_i\Phi(c_i)\right\|, \tag{4}$$

where $\Theta \subset \mathbb{R}^d$ is a compact set. As a heuristic to address the non-convex moment-matching problem, Keriven et al. (2017) proposed Compressive Learning-OMP (CL-OMPR), which is an adaptation of *Orthonormal Matching Pursuit* (OMP), a widely used algorithm in the field of sparse recovery (Mallat & Zhang, 1993; Pati et al., 1993). In a nutshell, CL-OMPR minimizes the residual $\|z_\mathcal{X} - \sum_{i=1}^k \alpha_i\Phi(c_i)\|$ by adding "atoms" $\Phi(c_i)$ in a greedy fashion; see Algorithm 1 for a simplified version of CL-OMPR and Algorithm 3 in Appendix A for the details. The definition of CL-OMPR was indeed inspired by a variant of OMP, namely OMP *with replacement* (OMPR), and adapted to the continuous setting with gradient descent steps. This is similar in spirit to Sliding Franke-Wolfe (Denoyelle et al., 2020), a continuous adaptation of Franke-Wolfe which is itself related to OMP, see e.g. Cherfaoui et al. (2019). Despite a promising behaviour in several empirical proofs of concept, CL-OMPR is only a heuristic and our first main contribution in the next section is to highlight its weaknesses, before using the resulting diagnoses to propose a new decoder and demonstrating the resulting improved performance and robustness.

---

**Data:** Sketch $z_\mathcal{X}$, sketching operator $\mathcal{A}$, parameters $k,T \geq k$, domain $\Theta$
**Result:** Set of centroids $C$, vector of weights $\boldsymbol{\alpha}$
$r \leftarrow z_\mathcal{X}$; $C \leftarrow \emptyset$;
**for** $i = 1 \ldots T$ **do**

    **Step 1:** *Find a new centroid and expand the support $C$*

        $f_r(c) := \mathfrak{Re}\langle r, \mathcal{A}\delta_c\rangle$                          $// \langle z, z'\rangle := \sum_{j=1}^m z_j \overline{z'_j}, \ \ \forall z, z' \in \mathbb{C}^m$

        $c \leftarrow \arg\max_{c\in\Theta} f_r(c)$

        $C \leftarrow C \cup \{c\}$

    **end**

    **Step 2:** *Reduce the support $C$ by Hard Thresholding when $i > k$*

    **Step 3:** *Project to find $\boldsymbol{\alpha}$ : $\boldsymbol{\alpha} \leftarrow \arg\min_{\boldsymbol{\alpha}\geq 0} \|z_\mathcal{X} - \sum_{i=1}^{|C|}\alpha_i\Phi(c_i)\|$*

    **Step 4:** *Fine tuning by gradient descent steps: $C, \boldsymbol{\alpha} \leftarrow \arg\min_{C\subset\Theta,\boldsymbol{\alpha}\geq 0} \|z_\mathcal{X} - \sum_{i=1}^{|C|}\alpha_i\Phi(c_i)\|$*

    **Step 5:** *Update the residual: $r \leftarrow z_\mathcal{X} - \sum_{i=1}^{|C|}\alpha_i\Phi(c_i)$*

**end**

**Algorithm 1:** CL-OMPR

---

## 3  Illustrating and diagnosing failures of CL-OMPR

Since its introduction by Keriven et al. (2017), CL-OMPR has been popular among methods that cluster using the sketch in equation 1 (Chatalic et al., 2018; 2022). In this section, we show how CL-OMPR is in fact not robust in the task of the recovery of centroids. To simplify the analysis, since the goal is to show that failures can happen *even in favorable scenarios*, we consider a setting where: i) the data is drawn from a mixture of well-separated Gaussians, ii) the dataset size $N$ is very large, and iii) the sketch size $m$ is very large. We show through a numerical experiment, that CL-OMPR fails to accurately reconstruct the centroids even in these optimal conditions.

### 3.1  A numerical experiment

We consider a dataset $\mathcal{X} = \{x_1, \ldots, x_N\} \subset \mathbb{R}^2$, where $N = 100000$ and the $x_i$ are i.i.d. draws from a mixture of isotropic Gaussians $\sum_{i=1}^k \alpha_i \mathcal{N}(c_i, \Sigma_i)$, where $k = 3$, $\alpha_1 = \alpha_2 = \alpha_3 = 1/3$, $c_1, c_2, c_3 \in \mathbb{R}^2$ and $\Sigma_1 = \cdots = \Sigma_k = \sigma_\mathcal{X}^2 \mathbb{I}_2 \in \mathbb{R}^{2\times 2}$ as shown in Figure 1b. We consider a sketching operator defined through random Fourier features associated to i.i.d. Gaussian frequencies $\omega_1, \ldots, \omega_M$ drawn from $\mathcal{N}(0, \sigma^{-2}\mathbb{I}_2)$ with

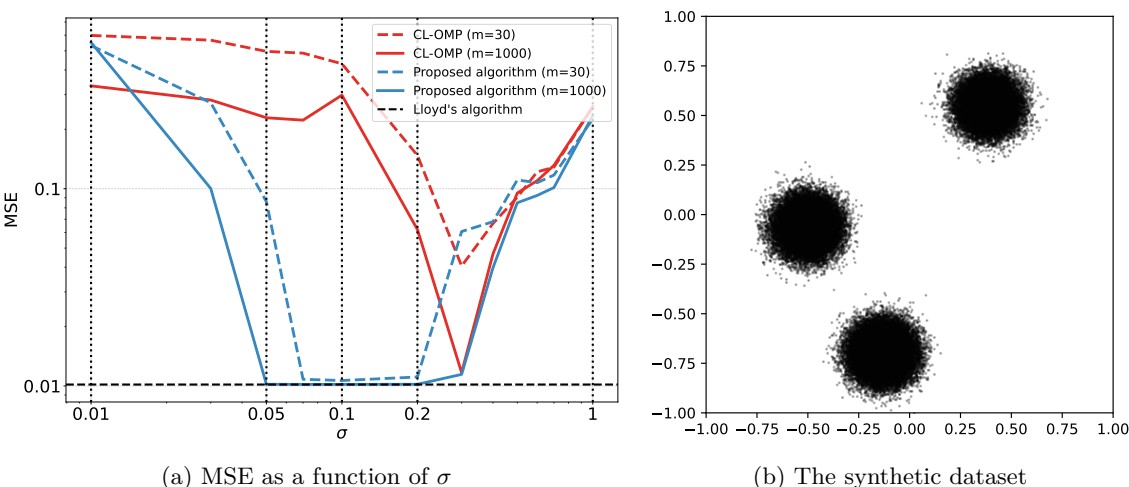

(a) MSE as a function of $\sigma$          (b) The synthetic dataset

Figure 1: Average MSE of Lloyd's algorithm, CL-OMPR for two sketch sizes, and the proposed Algorithm 2 on a synthetic dataset with three well-separated clusters in dimension 2. For well chosen values of $\sigma$ the MSE is of the order of 0.01, to be compared with inter-cluster squared distances of the order of 0.25, and intra-cluster variances of the order of 0.05.

$\sigma > 0$. Recall that, in the context of k-means clustering, the mean squares error (MSE) of a set of centroids $\mathcal{C} = \{c_1, \ldots, c_k\}$ is defined by

$$\mathrm{MSE}(\mathcal{C}; \mathcal{X}) := \frac{1}{N} \sum_{n=1}^{N} \min_{i \in [k]} \|x_n - c_i\|^2. \tag{5}$$

This is the quantity directly optimized by Lloyd's algorithm, while CL-OMPR instead addresses the optimization problem equation 4 as a proxy, both with constraint $c_i \in \Theta := [-1, 1]^2$ here. Figure 1a compares the MSE of Lloyd's algorithm, compressive clustering using CL-OMPR, and compressive clustering using Algorithm 2 for sketch sizes $m \in \{30, 1000\}$, averaged over 50 realizations of the sketching operator, as a function of the "bandwidth" $\sigma$ (of random Fourier features). As in previous work, we observe that the performance of CL-OMPR improves for large values of $m$. Moreover, we observe that, for $m = 1000$, the performance of CL-OMPR can be close to that of Lloyd-max, but the range of $\sigma$ for which this is the case is narrow ($\sigma \approx 0.3$), indicating that the parameter $\sigma$ of the sketched clustering pipeline is hard to tune. On the contrary, the performance of Algorithm 2 is the *same* as Lloyd's algorithm in a wider range of the bandwidth ($\sigma \in [0.03, 0.3]$), even at moderate sketch sizes $m = 30$ for which CL-OMPR is unable to approach this performance for any value of $\sigma$.

### 3.2 Diagnoses via links with kernel density estimation

As we now show, compressive clustering bears strong links with kernel density estimation, and some of the difficulties in tuning $\sigma$ observed above are reminiscent of classical kernel size tuning issues (Chen, 2017). We also highlight more specific difficulties of CL-OMPR as a decoder, which will help us design a better decoder (Algorithm 2) in the next section.

To relate compressive clustering to kernel density estimation, we examine the *correlation function*, which appears in Step 1 of CL-OMPR (Algorithm 1).

**Definition 1.** *Given a sketching operator $\mathcal{A} : \mathcal{P}(\mathbb{R}^d) \to \mathbb{C}^m$, and given $r \in \mathbb{C}^m$, define the* correlation function $f_r : \mathbb{R}^d \to \mathbb{R}$ *associated to $r$ and $\mathcal{A}$ as*

$$f_r(x) := \mathfrak{Re}\langle r, \mathcal{A}\delta_x \rangle. \tag{6}$$

Figure 2 shows the correlation function $f_{z_{\mathcal{X}}}$ associated to $r = z_{\mathcal{X}} = \mathcal{A}\pi_N$, in the first step of CL-OMPR, with the sketching operator defined in Section 3.1 for $m \in \{100, 1000\}$, and $\sigma \in \{0.01, 0.05, 0.1, 0.2, 1\}$ where

$\pi_N = \sum_{i=1}^N \delta_{x_i}/N$ corresponds to the dataset represented by Figure 1b. We observe that, for $\sigma = 1$, the correlation function has a single global maximum, which does not reflect the dataset's "clustered" geometry. As for $\sigma \in \{0.1, 0.2\}$, the correlation function exhibits multiple local maxima, the most significant ones being aligned with the cluster locations: the correlation function retains the dataset's geometry. In the case $\sigma = 0.05$, the correlation function continues to depict the dataset's geometry while introducing a number of spurious local maxima, especially for a small sketch size $m = 100$. Finally, for $\sigma = 0.01$, the correlation function is heavily distorted by 'noise' especially for $m = 100$. In light of these observations, we expect that recovering the clusters from the sketch will be infeasible for high and small values of $\sigma$, whereas it might be possible but poses challenges from an optimization perspective for intermediate $\sigma$'s.

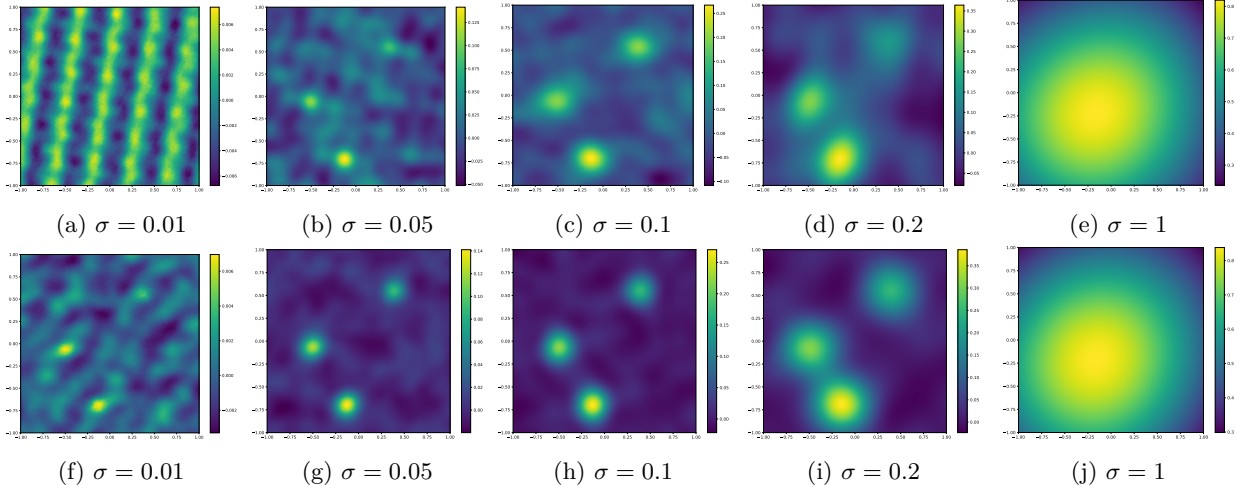

Figure 2: The correlation function for $m = 100$ (top) versus $m = 1000$ (bottom).

An important fact is that the correlation function approximates the kernel density estimator

$$\hat{f}_{\text{KDE}}(x) := \frac{1}{N} \sum_{i=1}^N \kappa_\sigma(x, x_i), \tag{7}$$

where $\kappa_\sigma(x, y) = \mathbb{E}_{\omega \sim \mathcal{N}(0, \sigma^{-2}\mathbb{I}_d)} \mathfrak{Re} \langle \mathcal{A}\delta_x, \mathcal{A}\delta_y \rangle$. Indeed, if $z_\mathcal{X} = \mathcal{A}\pi_N$, then

$$\mathbb{E}_{\omega \sim \mathcal{N}(0, \sigma^{-2}\mathbb{I}_d)} f_{z_\mathcal{X}}(x) = \frac{1}{N} \sum_{i=1}^N \mathbb{E}_{\omega \sim \mathcal{N}(0, \sigma^{-2}\mathbb{I}_d)} \mathfrak{Re} \langle \mathcal{A}x, \mathcal{A}x_i \rangle = \hat{f}_{\text{KDE}}(x). \tag{8}$$

As a consequence, step 1 in CL-OMPR (Algorithm 1) can be seen as a search for local maxima of $\hat{f}_{\text{KDE}}$ through its approximation $f_{z_\mathcal{X}}$. In other words, compressive clustering is related to density-based clustering methods, to which it brings the ability to work in memory-constrained scenarios, with distributed implementations, in streaming contexts, or with privacy constraints.

**Classical difficulties of density-based clustering.** Density-based clustering methods are attractive thanks to their capacity to identify clusters of arbitrary shapes: they define clusters as high density regions in the feature space (Fukunaga & Hostetler, 1975). Thus, some of the difficulties that the practitioner of compressive clustering may face are inherent to density-based clustering methods such as the selection of the kernel or the selection of the parameter of the kernel (Comaniciu & Meer, 2002): a very large value of $\sigma$ yields a smeared kernel and thus a smooth KDE for which the whole dataset belongs to a single cluster, while a very low value of $\sigma$ yields a trivial KDE where every point in the dataset belongs to its own cluster.

**Specific difficulties of CL-OMPR.** In addition, CL-OMPR (Algorithm 1) comes with its own difficulties. First, Step 1 is performed by randomly initializing a candidate centroid $c$ and performing gradient ascent on $f_r$. Such a gradient ascent can fail for several reasons:

- If the data is well clustered, then gradients are vanishingly small beyond a distance of roughly $\sigma$ away from the true cluster centroids. Gradient ascent will essentially not move the randomly initialized centroid $c$. This is illustrated on Figure 3 in Section 4.1.

- For moderate sketch sizes $m$, spurious local maxima of $f_{z_\mathcal{X}}$ appear due to Gibbs-like phenomena: the approximation of the smooth $\hat{f}_{\text{KDE}}$ of (6) by $f_{z_X}$ gives rise to oscillations, see Figure 2.

Second, the residual is updated in Step 5 by removing from $z_X$ the sketch of an estimated $|C|$-mixture of Diracs corresponding to the term $\sum_{i=1}^{|C|} \alpha_i \Phi(c_i) = \mathcal{A} \sum_{i=1}^{|C|} \alpha_i \delta_{c_i}$; see equation 3. In the most favorable case where each cluster is very localized, this is indeed likely to lead to a new correlation function $f_r$ with no local maximum around each of the already found centroids. However, the picture is quite different in the more realistic case where clusters have non-negligible (and possibly non-isotropic) intra-cluster variance. In the latter case, the correlation function $f_r$ of the residual updated in CL-OMPR indeed often still has a local maximum quite close to an already found centroid, leading the algorithm to repeatedly select the same dominant clusters and missing the other ones. This is illustrated in Figure 4 in Section 4.2, where we propose a fix. Finally, we observed that due to Step 2 of Algorithm 1, after first increasing the number of selected centroids from 1 to $k$, CL-OMPR repeats $T - k$ times the addition of a new one followed by the removal of one, thus never exploring more than $k + 1$ candidates at a time. As we will, a variant allowing more exploration is beneficial.

## 4 Towards a robust decoder for compressive clustering

In this section, we propose a decoder that overcome the limitations of CL-OMPR mentioned in the end of Section 3.2. For this, we introduce three ingredients: i) several approches to handle the non-convex optimization problem that appears in Step 1 of CL-OMPR, which consists in seeking the centroids among the local maxima of the correlation function $f_{z_\mathcal{X}}$, ii) reducing the support $C$ through hard-thresholding is executed after selecting all the candidate centroids, iii) allowing to fit a $|C|$-mixture of *Gaussians* instead of *Diracs* to take into consideration clusters with non-negligible (and possibly nonisotropic) intra-cluster (co)variance: considering the selected $c$ from Step 1 as the mean of a new Gaussian, this requires estimating the covariance matrix of this added Gaussian.

Algorithm 2 proceeds in two steps. The first one, which is reminiscent of Orthogonal Matching Pursuit (Pati et al., 1993; Mallat & Zhang, 1993), consists in seeking an initial support $\tilde{C} \subset \Theta$ of $T \geq k$ candidate centroids. This is achieved by iteratively seeking a candidate point $c_i$ that locally maximizes the correlation function $f_r$, then seeking the vector $\boldsymbol{\alpha} \in \mathbb{R}_+^t$ that minimizes a residual, which expression depends on the fitted model: a mixture of Diracs, or a mixture of Gaussians. The latter requires the estimation of the local covariance matrix $\Sigma_i$ using a procedure `EstimateSigma`, that we will motivate in Section 4.2 and detail in Appendix B. This procedure either outputs a valid (i.e., positive definite) covariance matrix, or zero, in which case the corresponding component is set to a Dirac. Observe that updating the residual in every iteration is crucial (but not sufficient – this motivates to consider Gaussian mixtures) in order to obtain candidate centroids that are different at each iteration. This step depends on the fitted model. When fitting a mixture of Gaussians, updating the residual boils down to calculating the terms $\mathcal{A}\mathcal{N}(c_i, \Sigma_i)$, which are equal to the characteristic functions of the corresponding Gaussians when $\mathcal{A}$ is build using random Fourier features (Keriven, 2017, Equation 5.7). The second step consists in pruning the support to $C$ to obtain the targeted number $k$ of centroids.

We next discuss the choice of the procedures `GetLocalMaximum` and `EstimateSigma`.

### 4.1 Detecting local maxima of the correlation function

We describe in this section, three algorithms to implement `GetLocalMaximum` in Step 1 of Algorithm 2. The pros and cons of these variants are discussed theoretically and will be assessed numerically in Section 5.

**Discretized approach:** The first option consists in replacing the optimization problem $\max_{c \in \Theta} f_r(c)$ by $\max_{c \in \tilde{\Theta}} f_r(c)$, where $\tilde{\Theta} \subset \Theta$ is a (finite) discretization of $\Theta$. This approach is recurrent in the literature of

**Data:** Sketch $z_\mathcal{X}$, sketching operator $\mathcal{A}$, number of centroids $k$, number of atoms $T \geq k$, fitted model 'model', a domain $\Theta \subset \mathbb{R}^d$

**Result:** The set of centroids $C = \{c_1, \ldots, c_k\}$, the vector of weights $\boldsymbol{\alpha} = (\alpha_1, \ldots, \alpha_k)$

$r \leftarrow z_\mathcal{X}; \quad C \leftarrow \emptyset;$

**Step 1:** *Look for an initial support*

> **for** $i = 1, \ldots, T$ **do**
>> $f_r(x) \leftarrow \mathfrak{Re}\langle r, \mathcal{A}\delta_x \rangle$
>> $c_i \leftarrow \texttt{GetLocalMaximum}(f_r; \Theta)$ //Find a new centroid
>> $C \leftarrow C \cup \{c_i\}$ //Expand the support
>> **if** model $=$ Dirac **then**
>>> $\boldsymbol{\alpha} \leftarrow \arg\min_{\boldsymbol{\alpha} \in \mathbb{R}_+^{|C|}} \|z_\mathcal{X} - \sum_{j=1}^{|C|} \alpha_j \mathcal{A}\delta_{c_j}\|$ //Project to find the weights
>>> $r \leftarrow z_\mathcal{X} - \sum_{j=1}^{|C|} \alpha_j \mathcal{A}\delta_{c_j}$ //Update the residual
>> **end**
>> **else if** model $=$ Gaussian **then**
>>> $\Sigma_i \leftarrow \texttt{EstimateSigma}(f_{z_\mathcal{X}}, c_i)$//Estimate the covariance matrix
>>> **if** $\Sigma_i \equiv 0$ **then**
>>>> $\pi_i \leftarrow \mathcal{N}(c_i, \Sigma_i)$//Define new Gaussian component
>>> **end**
>>> **else if** $\Sigma_i \neq 0$ **then**
>>>> $\pi_i \leftarrow \delta_{c_i}$ //Revert to Dirac component
>>> $\boldsymbol{\alpha} \leftarrow \arg\min_{\boldsymbol{\alpha} \in \mathbb{R}_+^{|C|}} \|z_\mathcal{X} - \sum_{j=1}^{|C|} \alpha_j \mathcal{A}\pi_j\|$ //Find the weights
>>> $r \leftarrow z_\mathcal{X} - \sum_{i=1}^{|C|} \alpha_i \mathcal{A}\mathcal{N}(c_i, \Sigma_i)$ //Update the residual
>> **end**

**Step 2:** *Reduce the initial support when $T > k$*

> **if** $|C| > k$ **then**
>> Select indices $i_1, \ldots, i_k$ of the largest $k$ elements of $\boldsymbol{\alpha}$: $\alpha_{i_1}, \ldots, \alpha_{i_k}$
>> $C \leftarrow \{c_{i_1}, \ldots, c_{i_k}\}$ //Reduce the support
>> $\boldsymbol{\alpha} \leftarrow (\alpha_{i_1}, \ldots, \alpha_{i_k})$ //Reduce $\boldsymbol{\alpha}$
> **end**

**Algorithm 2:** Proposed decoder

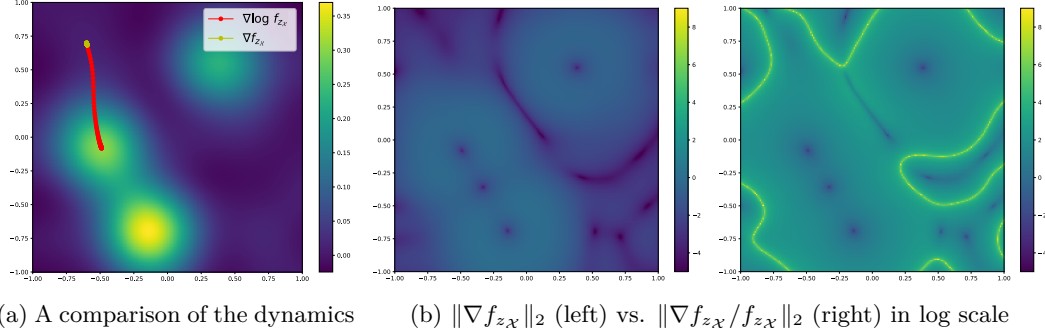

(a) A comparison of the dynamics      (b) $\|\nabla f_{z_\mathcal{X}}\|_2$ (left) vs. $\|\nabla f_{z_\mathcal{X}} / f_{z_\mathcal{X}}\|_2$ (right) in log scale

Figure 3: A comparison between plain gradient ascent and sketched mean shift in the identification of a local maximum of $f_{z_\mathcal{X}}$; dynamic ranges of $\|\nabla f_{z_\mathcal{X}}\|_2$ and $\|\nabla f_{z_\mathcal{X}} / f_{z_\mathcal{X}}\|_2$.

superresolution (Tang et al., 2013; Traonmilin et al., 2020), where $\Theta$ is usually a subset of a low-dimensional vector space. We show later that this approach yields significant improvement over CL-OMPR. However, in high-dimensional settings $\Theta \subset \mathbb{R}^d$, the cardinality of $\tilde{\Theta}$ must grow exponentially with the dimension $d$.

**Sketched mean shift approach:** Instead of discretizing, we may use a gradient-based method to identify local maxima of the correlation function. In this context, a simple gradient ascent is ineffective in practice

because the gradient $\nabla f_r$ is vanishingly weak beyond the immediate vicinity of some points, resulting in a slow gradient dynamic; see Figure 3 for an illustration in the case of $r = z_{\mathcal{X}}$.

As an alternative, we propose to perform reweighted gradient ascent iterations

$$c \leftarrow \Pi_{\Theta}(c + \frac{\eta}{|f_r(c)|}\nabla f_r(c)), \tag{9}$$

where $\Pi_{\Theta}$ is the projection onto $\Theta$ with respect to the Euclidean distance. This boils down to the ascent algorithm based on the gradient of $\log f_r$, when $f_r$ takes positive values. As illustrated in Figure 3a in the case of the correlation function $f_{z_{\mathcal{X}}}$, this algorithm exhibits improved dynamics compared to plain ascent of the gradient of $f_r$: the trajectory from the same starting point hardly changes when using plain gradient ascent iterations, but it does reach a centroid when performing the reweighted gradient ascent of equation 9. Figure 3b compares the norm of $\nabla f_{z_{\mathcal{X}}}$ to the norm of $\nabla f_{z_{\mathcal{X}}}/f_{z_{\mathcal{X}}}$: we observe that the former is very small in most regions while the latter reaches much larger values in general, except nearby stationary points where it vanishes. Now, equation 9 is reminiscent of the mean shift algorithm (Fukunaga & Hostetler, 1975; Cheng, 1995) mentioned in Section 3.2. The latter boils down to a gradient ascent applied to the logarithm of the KDE, defined by equation 7, when $\kappa_{\sigma}$ is the Gaussian kernel, hence the name *sketched mean shift*. Still, the usual mean shift algorithm requires access to the whole dataset $\mathcal{X}$ in every iteration, while equation 9 doesn't, thanks to the very principle of the sketching approach which summarizes the dataset into $z_{\mathcal{X}}$. Observe that making use of the iteration equation 9 requires that the function $f_r$ does not vanish on the whole path of optimization. This property was empirically observed in the numerical simulations presented in Section 5.

The sketched mean shift algorithm might still converge to a spurious local maximum, particularly when the sketch size $m$ is low. Thus, the best output of sketched mean shift over $L$ random initializations of $c$ is computed. As we will see in the experiments Section 5, this approach is more efficient than discretization.

## 4.2 The importance of the fitted model

Figure 4 illustrates the benefit of fitting a $k$-mixture of Gaussians instead of a $k$-mixture of Diracs during the iterations of Algorithm 2: with mixtures of Dirac, once the residual $r$ is updated, the corresponding correlation function $f_r$ still has a (high) local maximum close to the previously found centroid, leading the algorithm to select a nearly identical cluster. This no longer happens when fitting a mixture of Gaussians, with properly chosen covariance. The implementation of this in Algorithm 2 requires to estimate the local covariance matrix once a centroid $c$ is selected in Step 1. In Appendix B, we propose an estimator of this matrix. As we shall see in Section 5, this allows to robustify further the decoder for lower values $\sigma$ than if we simply fit $k$-mixture of Diracs. The theoretical study of this estimator, especially in high-dimensional domains, is left for a future work.

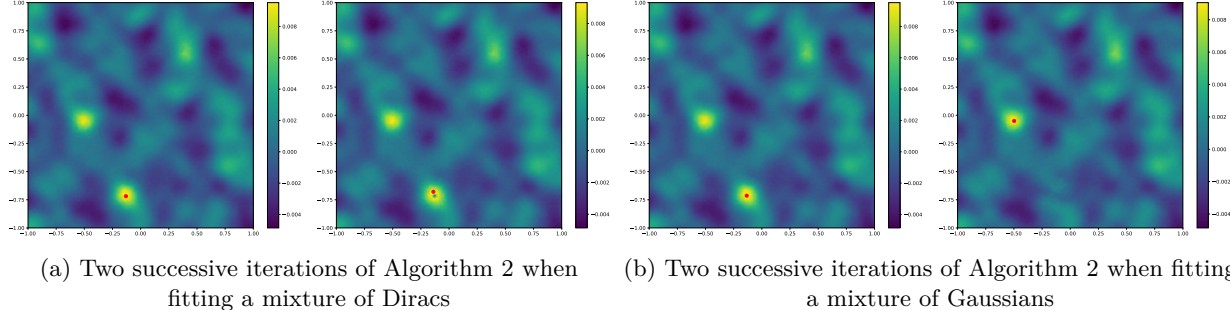

(a) Two successive iterations of Algorithm 2 when fitting a mixture of Diracs

(b) Two successive iterations of Algorithm 2 when fitting a mixture of Gaussians

Figure 4: The correlation function $f_r$ associated to the residual $r$ at the first and second iteration of Algorithm 2 using two fitted models: mixture of Diracs and mixture of Gaussians. The red points correspond to the selected centroids.

# 5 Numerical simulations

In this section we present numerical experiments that illustrate the performance of the newly proposed decoder. In Section 5.1 we conduct numerical experiments on synthetic datasets to study the performance of the decoder in high-dimensional settings. In Section 5.2, we investigate the importance of the fitted model using synthetic datasets. Finally, in Section 5.3, we perform the experiments on real datasets.

## 5.1 The discretized approach compared to the sketched mean shift approach

In this section, we compare the two variants of Algorithm 2 proposed in Section 4.1. For this purpose, we conduct numerical simulations on a synthetic dataset $\mathcal{X} \subset \mathbb{R}^6$ made of three clusters: the dataset $\mathcal{X} = \{x_1, \ldots, x_N\}$ contains $N = 100000$ vectors in $\mathbb{R}^6$ which are obtained by i.i.d. sampling from a mixture of isotropic Gaussians $\sum_{i=1}^{k} \alpha_i \mathcal{N}(c_i, \Sigma_i)$, where $k = 3$, $\alpha_1 = \alpha_2 = \alpha_3 = 1/3$, $c_1, c_2, c_3 \in \mathbb{R}^2$, and $\Sigma_1 = \cdots = \Sigma_k = \sigma_{\mathcal{X}}^2 \mathbb{I}_2 \in \mathbb{R}^{2 \times 2}$. The centroids $c_i$ and variance $\sigma_{\mathcal{X}}^2$ were chosen using a function implemented in the Python package Pycle[2] to ensure enough separation while essentially fitting the dataset [3] in $[-1, 1]^6$. For all algorithms we use as a domain $\Theta$ the hypercube $\Theta = [-1, 1]^6$.

Performance is defined *relative to the performance of Lloyd's algorithm*, i.e., using a normalized version of the MSE called the relative squared error (RSE) and widely used in previous work on compressive clustering. The RSE is defined as

$$\mathrm{RSE}(\mathcal{C}; \mathcal{X}) := \frac{\mathrm{MSE}(\mathcal{C}; \mathcal{X})}{\mathrm{MSE}(\mathcal{C}_{\mathrm{Lloyd}}; \mathcal{X})}, \tag{10}$$

where $\mathrm{MSE}(\mathcal{C}; \mathcal{X})$ is given by equation 5 and $\mathcal{C}_{\mathrm{Lloyd}}$ is the configuration of centroids given by Lloyd's algorithm (best of 5 runs with different centroid seeds).

Figure 5a compares the RSE of compressive clustering using CL-OMPR, compressive clustering using Algorithm 2 based on the discretized approach, and compressive clustering using Algorithm 2 based on the sketched mean shift approach. By definition, the RSE of Lloyd's algorithm is one.

For each compressive approach, the RSE is averaged on 50 realizations of the sketching operator and presented for two sketch sizes $m \in \{200, 1000\}$. For the three compressive algorithms, we take $T = 2k$. Moreover, for the two variants of Algorithm 2, we take $L = 10000$ random initializations that are i.i.d. samples from the uniform distribution on $\Theta = [-1, 1]^6$. We observe that both CL-OMPR and Algorithm 2 based on the discretized approach fail to match or even approach the performance of Lloyd's algorithm (i.e., to achieve an RSE close to one) for any bandwidth of $\sigma$, while the performance of Algorithm 2 based on the sketched mean shift approach is very similar to that of Lloyd's algorithm for $m = 200$, and matches it for $m = 1000$ in a range of bandwidths ($\sigma \in [0.1, 0.3]$). In other words, CL-OMPR fails to capture the local maxima of $f_r$. The use of weighted gradient descent as in equation 9 in the second variant allows to better capture the local maxima, hence Algorithm 2 based on the sketched mean shift approach outperforms Algorithm 2 based on the discretized approach. As a result from now on when considering Algorithm 2 we systematically focus on the sketched mean shift approach.

Now, we examine how the number of initializations $L$ influences the performance of Algorithm 2. Figure 5b compares the RSE of compressive clustering using CL-OMPR, and compressive clustering using Algorithm 2 based on the sketched mean shift approach associated to various values of $L$. We observe that the range of the bandwidth $\sigma$ for which Algorithm 2 matches Lloyd's algorithm (i.e., achieves an RSE close to one) increases with $L$ for both considered values of $m$. In particular, we observe that the performance of Algorithm 2 for large values of $\sigma$ ($\sigma > 0.3$ when $m = 1000$ and $\sigma > 0.4$ when $m = 200$) does not depend on $L$, while it is highly dependent on the value of $L$ for low values of $\sigma$. In light of the 2D illustrations of Section 3, this is likely due to the existence of spurious local maxima for low values of the bandwidth $\sigma$. Indeed, spurious local maxima proliferate as $\sigma$ gets smaller; see Figure 2 for an illustration. This numerical experiment shows that the recovery of the centroids from the sketch using Algorithm 2 is possible even for low values of $\sigma$ by increasing $L$. Figure 6 shows the relative squared error (RSE) of compressive clustering using CL-OMPR,

---

[2]https://github.com/schellekensv/pycle
[3]The dataset and the code to generate a similar one can be downloaded from https://openreview.net/forum?id=6rWuWbVmgz.

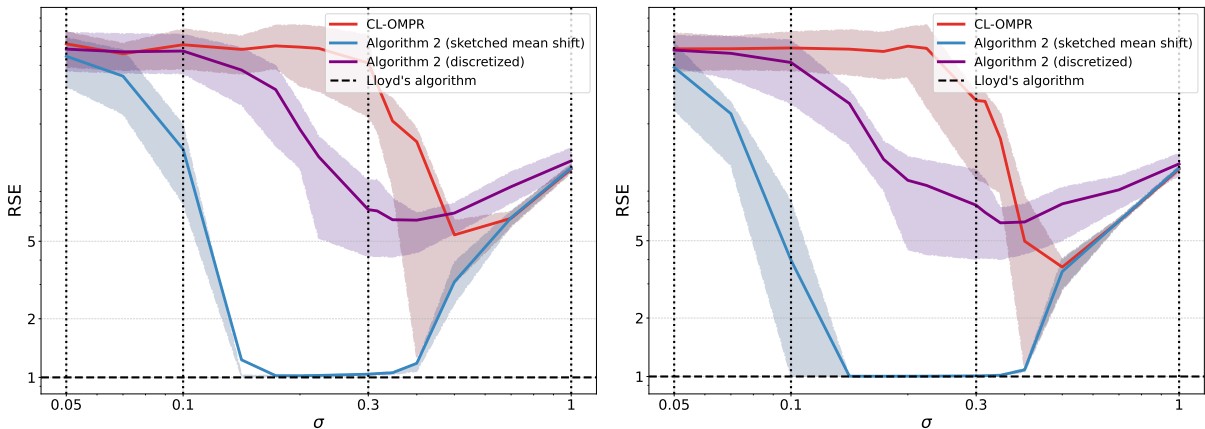

(a) Average RSE of CL-OMPR for two sketch sizes (left: $m = 200$; right: $m = 1000$), and Algorithm 2 with two approaches (discretized and sketched mean shift, $L = 1000$).

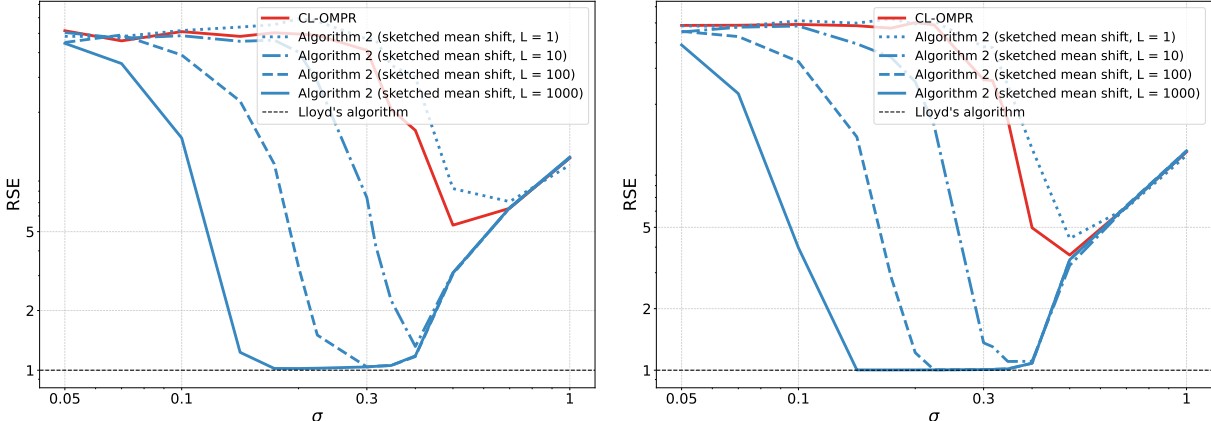

(b) Average RSE of CL-OMPR for two sketch sizes (left: $m = 200$; right: $m = 1000$), and Algorithm 2 using sketched mean shift for $L \in \{10, 100, 1000\}$.

Figure 5: Comparison of CL-OMPR and Algorithm 2 with three synthetic clusters in $[-1, 1]^6$

Algorithm 2 based on the sketched mean shift approach (with three values of $L$: $L = 10$, $L = 100$ and $L = 1000$), using 50 realizations of the sketching operator and presented for various values of the sketch size $m$ and the bandwidth parameter $\sigma$.

We observe that the range of the bandwidth $\sigma$ for which Algorithm 2 matches the performance of Lloyd's algorithm (this corresponds to reaching an RSE close to 1) increases with $m$ and $L$. More precisely, as $L$ gets larger, the performance of Algorithm 2 improves for $\sigma \leq 0.3$, yet without an improvement for larger values of $\sigma$ ($\sigma \geq 0.5$). In comparison, the average RSE of CL-OMPR stays above 2 for every value of the sketch size $m$ and the bandwidth $\sigma$.[4]

**Remark 1.** *All experiments in the main body of this paper are conducted in moderate dimension $d \leq 10$, which is pretty much the standard dimension of state of the art proofs of concept in sketched clustering. While Algorithm 2 is designed to overcome the important issues of CL-OMPR scrutinized in this paper, we do not expect this algorithm to be particularly good at overcoming issues related to a curse of dimension. In particular, it is expected that in high dimension, success will require an exponential growth of $L$. This is empirically confirmed with additional experiments in Appendix C.1. Another algorithmic challenge left to*

---

[4]We use in Figure 6a a threshold (RSE > 5) different than the threshold (RSE > 2) used in Figures 6b to 6d.

*future work is to handle large numbers K of clusters, as CL-OMPR is known to badly scale with K (Keriven et al., 2017).*

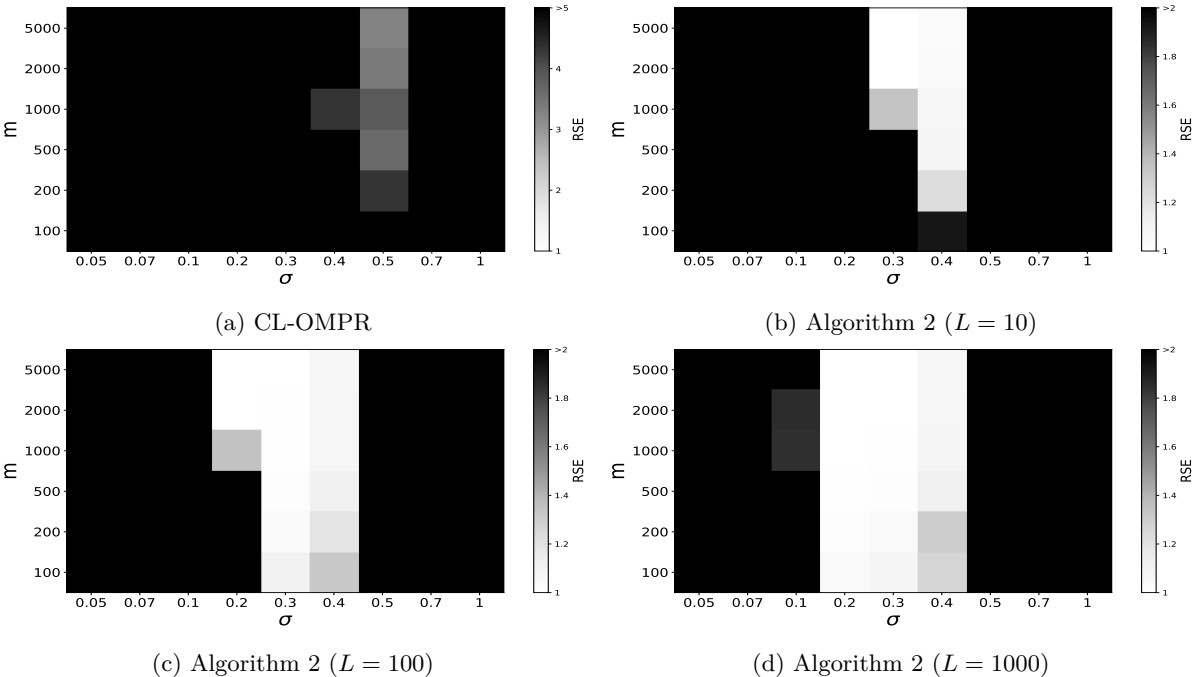

(a) CL-OMPR

(b) Algorithm 2 ($L = 10$)

(c) Algorithm 2 ($L = 100$)

(d) Algorithm 2 ($L = 1000$)

Figure 6: Comparison of CL-OMPR and Algorithm 2 on three *synthetic* clusters in $[-1, 1]^6$

## 5.2 On the importance of the fitted model

In this section, we investigate the importance of the fitted model. For this purpose, we compare fitting a $k$-mixture of Gaussians instead of a $k$-mixture of Diracs on two synthetic datasets. The first one is the one considered in Section 3, while the second one is formed by 3 clusters in $[-1, 1]^2$, that are not well separated; see Figure 7b. We compare Algorithm 2 using fitting with $k$-mixture of Diracs to Algorithm 2 using fitting with $k$-mixture of Gaussians. In the latter case, the local covariance matrix is estimated as detailed in Appendix B.

For each dataset, we use as a baseline the best out of 5 replicas (with different centroid seeds) of Lloyd's algorithm. For each instance of Algorithm 2 (with Diracs, or with Gaussians; both using the sketched mean shift approach with $L = 1000$), and each considered value of the sketch size $m$ and the bandwidth parameter $\sigma$, we compute the average RSE (cf equation 10) over 10 realizations of the sketching operator. Figure 7 shows the obtained RSE for various values of the sketch size $m$ and the bandwidth $\sigma$.

In the case of the dataset represented by Figure 7a, we observe that using Gaussians instead of Diracs primarily enlarges the range of the bandwidth $\sigma$ for which Algorithm 2 matches Lloyd's performance (RSE close to one). For the the dataset represented by Figure 7b, we make several observations: i) when fitting mixtures of Diracs, the largest bandwidth where the perfomance of Lloyd's algorithm is matched is smaller than with the dataset of Figure 7a even for large values of the sketch size ($m \geq 1000$), this is due to the increased difficulty in separating the clusters; ii) fitting mixture of Gaussians improves the performance of Algorithm 2 for low values of $\sigma$: the smallest bandwidth where the perfomance of Lloyd's algorithm is matched is smaller than when fitting mixture of Diracs.

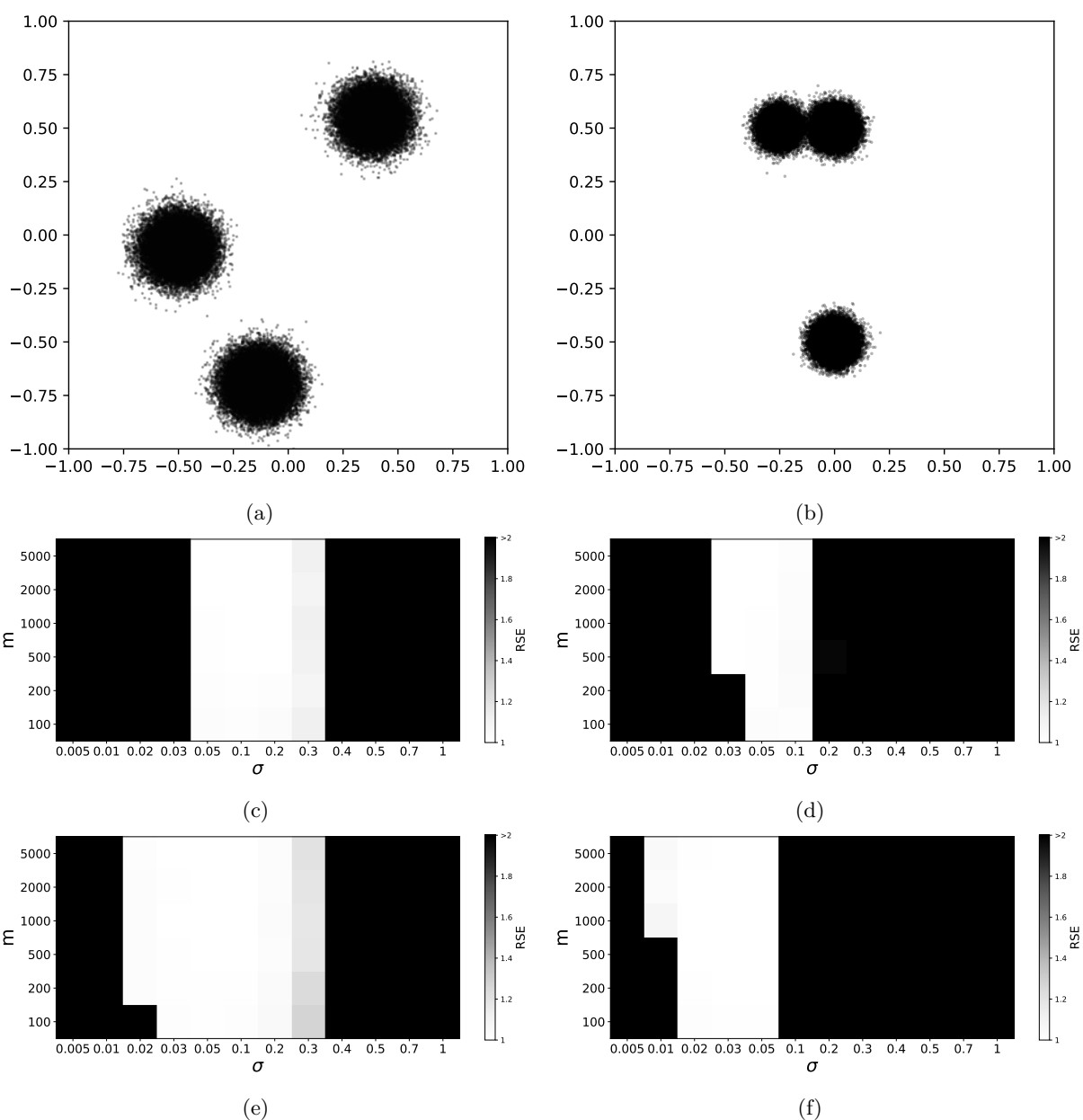

Figure 7: Performance of compressive clustering using Algorithm 2 on two datasets (left column: a well-separated dataset; right column: a dataset with closeby clusters). Top: representation of the datasets; middle: average RSE obtained for various sketch sizes $m$ and scales $\sigma$ using a *mixture of Diracs* model. Bottom: average RSE achieved using a *mixture of Gaussians* model

## 5.3 Experiments on MNIST

In this section, we investigate whether the observations of Section 5.2 hold for real datasets. For this purpose, we perform experiments on spectral features of the MNIST dataset[5], which consist of $N = 70000$ handwritten digits features with $k = 10$ classes. These spectral features are computed by taking the eigenvectors associated to the $d = 10$ smallest positive eigenvalues of the normalized Laplacian matrix associated to the nearest neighbors matrix (Muja & Lowe, 2009), associated to SIFT descriptors of each image (Vedaldi &

---

[5]Further experiments on CIFAR10 in Appendix C.2 display a similar behavior.

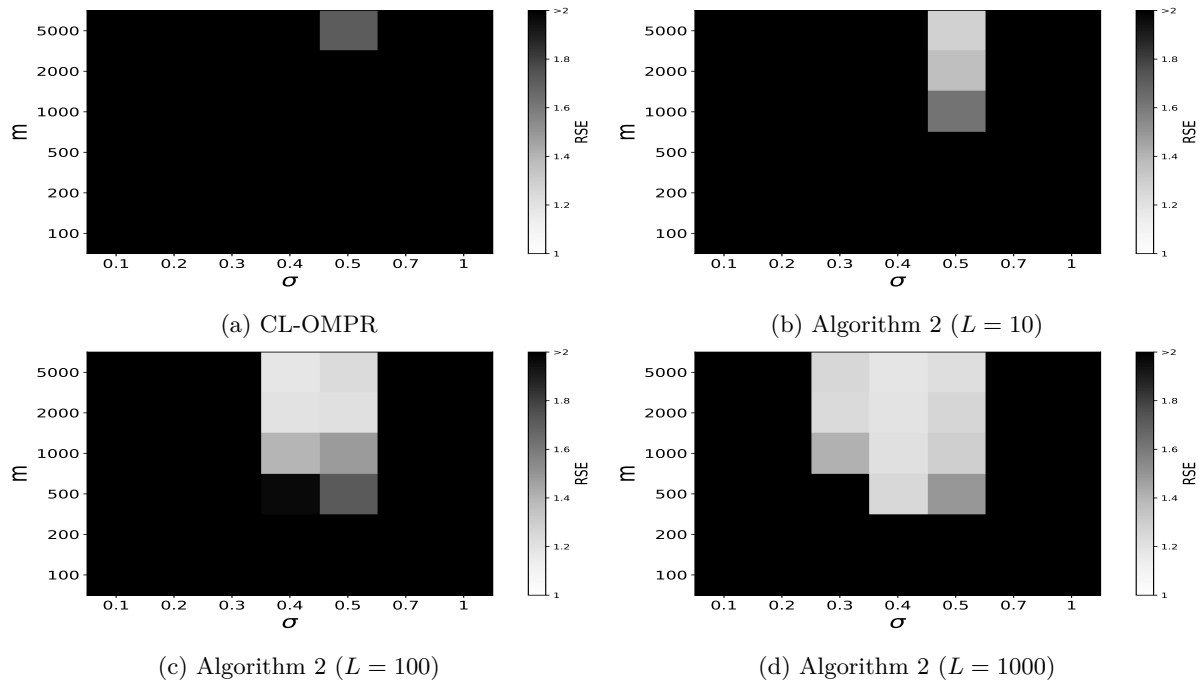

Figure 8: A comparison of CL-OMPR and Algorithm 2 on *spectral features of MNIST*

Fulkerson, 2010). The resulting matrix can be downloaded from https://gitlab.com/dzla/SpectralMNIST. Figure 8 shows the RSE, defined by equation 10 based on Lloyd's algorithm (best of 5 runs of Lloyd's algorithm with different centroid seeds), of compressive clustering using CL-OMPR, and Algorithm 2 with mixtures of Diracs[6] and the sketched mean shift approach (with three values of $L$: $L = 10$, $L = 100$, and $L = 1000$), based on 10 realizations of the sketching operator for various values of the sketch size $m$ and the bandwidth $\sigma$. Similarly to the experiment conducted above on the synthetic dataset, we observe that Algorithm 2 outperforms CL-OMPR for $\sigma \leq 0.5$, and get a performance close to Lloyd's algorithm (RSE $\leq 1.5$) on a range of $\sigma$ that increases with $m$ and $L$. In particular, for $L = 1000$, Algorithm 2 achieves an RSE smaller than 1.5 for $m = 500$ (only five times the number of degrees of freedom $kd = 100$ needed to describe $k = 10$ centroids in dimension $d = 10$), while the best achievable RSE by CL-OMPR is close to 2 and was achieved for a sketch size 10 times larger: $m = 5000$. For this dataset, the average running time for calculating the sketch is approximately 0.2 seconds, and the average running time for decoding using CL-OMPR is about 3 seconds. Meanwhile, the running time for decoding using Algorithm 9 depends linearly on the number of initializations $L$; for instance, it averages 12 seconds when $L = 10$. Improving the running time of Equation (9) is left to future work, as our primary effort in this paper in to demonstrate improved and robustified RSE performance over a wide range of values of $\sigma$ and $m$ compared to CL-OMPR.

## 6 Conclusion

By diagnosing CL-OMPR, we were able to propose a robustified decoder which significantly outperforms CL-OMPR on synthetic datasets. In particular, we demonstrated on a simple dataset that CL-OMPR fails, while our algorithm matches the performance of Lloyd's algorithm, even though (unlike Lloyd's algorithm) it only has access to a low-dimensional sketch, which dimension is independent of the size of the dataset. Moreover, compared to previous proofs of concept of compressive learning, our algorithm is able to extract clustering information from smaller sketches, therefore further reducing the needed memory footprint. Given the potential of compressive clustering (memory-constrained scenarios, with distributed implementations, in streaming contexts, or with privacy constraints), this work thus opens up unprecedented prospects for making

---

[6]Algorithm 2 with a Gaussian mixture led to worse results, likely due to limitations of the covariance estimation procedure of Appendix B in dimension $d = 10$. Improving this procedure for high dimensions is a challenge left to future work.

the full pipeline of this paradigm both competitive, easy to implement and easy to tune. Considering the choice of the bandwidth scale $\sigma$, which is currently something of an art, the robustness of our algorithm combined with recently proposed selection criteria (Giffon & Gribonval, 2022) open a promising avenue to design a fully turnkey pipeline for compressive clustering. The next upcoming challenge is of course to validate the versatility of the resulting approach on datasets living in high-dimensional domains, while controlling the number of required intializations $L$ of the sketched mean shift to control its computational complexity. To achieve this goal, leveraging alternative feature maps would be beneficial (Avron et al., 2017; Chatalic et al., 2022). On another vein, a promising research path involves estimating local covariance matrices from the sketch. Given the strong connection of our approach with the mean shift algorithm, for which the convergence was actively investigated recently (Li et al., 2007; Ghassabeh, 2015; Huang et al., 2018; Yamasaki & Tanaka, 2019; 2023), we aim to investigate the theoretical properties of sketched mean shift. Finally, the connection of the mean shift algorithm to EM algorithm (Carreira-Perpinan, 2007) suggests the possibility to extend our approach in parameter estimation of more general mixture models such as mixture of alpha-stable distributions (Keriven et al., 2018b).

## Acknowledgments

This project was supported by the AllegroAssai ANR project ANR-19-CHIA-0009. The authors thank the Blaise Pascal Center (CBP) for the computational means. It uses the SIDUS solution developed by Quemener & Corvellec (2013).

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

# A The CL-OMPR algorithm

**Data:** Sketch $z_{\mathcal{X}}$, sketching operator $\mathcal{A}$, parameter $k$, bounds $\boldsymbol{\ell}, \boldsymbol{u}$
**Result:** Centroids $C$, weights $\boldsymbol{\alpha}$
$r \leftarrow z_{\mathcal{X}}; C \leftarrow \emptyset;$
**for** $t = 1 \ldots 2k$ **do**

> **Step 1:** *Find a new centroid*
>
> > $\boldsymbol{c} \leftarrow \max_{\boldsymbol{c}} \left( \text{Re} \left\langle \frac{\mathcal{A}\delta_{\boldsymbol{c}}}{\|\mathcal{A}\delta_{\boldsymbol{c}}\|}, r \right\rangle, \boldsymbol{\ell}, \boldsymbol{u} \right)$
>
> **end**
> **Step 2:** *Expand support*
> > $C \leftarrow C \cup \{\boldsymbol{c}\}$
> **end**
> **Step 3:** *Enforce sparsity by Hard Thresholding when $t > k$*
> > **if** $|C| > k$ **then**
> > > $\boldsymbol{\beta} \leftarrow \arg\min_{\boldsymbol{\beta} \geq 0} \left\| z_{\mathcal{X}} - \sum_{i=1}^{|C|} \beta_i \frac{\mathcal{A}\delta_{\boldsymbol{c}_i}}{\|\mathcal{A}\delta_{\boldsymbol{c}_i}\|} \right\|$
> > > Select $k$ largest entries $\beta_{i_1}, \ldots, \beta_{i_k}$
> > > Reduce the support $C \leftarrow \{\boldsymbol{c}_{i_1}, \ldots, \boldsymbol{c}_{i_k}\}$
> > **end**
> **end**
> **Step 4:** *Project to find $\boldsymbol{\alpha}$*
> > $\boldsymbol{\alpha} \leftarrow \arg\min_{\boldsymbol{\alpha} \geq 0} \left\| z_{\mathcal{X}} - \sum_{i=1}^{|C|} \alpha_i \mathcal{A}\delta_{\boldsymbol{c}_i} \right\|$
> **end**
> **Step 5:** *Global gradient descent*
> > $C, \boldsymbol{\alpha} \leftarrow \min_{C, \boldsymbol{\alpha}} \left( \left\| z_{\mathcal{X}} - \sum_{i=1}^{|C|} \alpha_i \mathcal{A}\delta_{\boldsymbol{c}_i} \right\|, \boldsymbol{\ell}, \boldsymbol{u} \right)$
> **end**
> Update residual: $r \leftarrow z_{\mathcal{X}} - \sum_{i=1}^{|C|} \alpha_i \mathcal{A}\delta_{\boldsymbol{c}_i}$

**end**

**Algorithm 3:** CL-OMPR

# B An estimator of the local covariance matrix

Algorithm 2 allows to fit a $k$-mixture of Gaussians, which requires to estimate the local covariance matrix once a centroid $c$ is selected in Step 1. In this section, we propose an estimator of this matrix. This provides an implementation of `EstimateSigma` for Algorithm 2.

To begin with an intuition, consider the following setting: the $x_i$ are i.i.d. draws from a mixture of isotropic Gaussians $\sum_{i=1}^{k} \alpha_i \mathcal{N}(c_i, \Sigma_i)$, where $\alpha_1, \ldots, \alpha_k \in [0, 1]$ such that $\sum_{i=1}^{k} \alpha_i = 1$ and $c_1, \ldots, c_k \in \Theta \subset \mathbb{R}^d$ and $\Sigma_1, \ldots, \Sigma_k \in \mathbb{S}_d^{++}$ with $k \in \mathbb{N}^*$. We consider a sketching operator defined through random Fourier features associated to i.i.d. Gaussian frequencies $\omega_1, \ldots, \omega_M$ drawn from $\mathcal{N}(0, \sigma^{-2}\mathbb{I}_2)$, with $\sigma > 0$. As the number of samples $N$ goes to $+\infty$ the KDE converges to the kernel mean embedding $f_{\text{KME}}$ of the Gaussian mixture $\sum_{i=1}^{k} u_i \mathcal{N}(c_i, \Sigma_i)$ (see Section 3.1.2 in (Muandet et al., 2017)), which is given by (see Lemma 6.4.1 in (Keriven, 2017))

$$f_{\text{KME}}(x) := \sum_{i=1}^{k} u_i \exp\left( -\frac{1}{2}(x - c_i)^{\text{T}} (\Sigma_i + \sigma^2 \mathbb{I}_d)^{-1} (x - c_i) \right); x \in \mathbb{R}^d. \tag{11}$$

Now, when the clusters $c_1, \ldots, c_k$ are well-separated, we have for $i \in \{1, \ldots, k\}$

$$\log f_{\text{KME}} \approx_{x \to c_i} -\frac{1}{2}(x - c_i)^{\text{T}}(\Sigma_i + \sigma^2 \mathbb{I}_d)^{-1}(x - c_i) + \log u_i. \tag{12}$$

In other words, $-\log f_{\text{KME}}$ behaves locally as a quadratic function associated to the p.s.d. matrix $(\Sigma_i + \sigma^2 \mathbb{I}_d)^{-1}$. Thus, if $\hat{H}$ is an estimate of the Hessian of $-\log f_{\text{KME}}$ then $\hat{H}^{-1} - \sigma^2 \mathbb{I}_d$ is an estimate of $\Sigma_i$. Since an estimated covariance matrix must be p.s.d., if the matrix $\hat{H}^{-1} - \sigma^2 \mathbb{I}_d$ turns out to be *not* p.s.d., then we set as our estimate $\Sigma_i := 0$ to revert to a Dirac component, see Algorithm 2. Thus, to build an estimator of $\Sigma_i$ we need to estimate the Hessian of $-\log f_{\text{KME}}$.

The following estimator is simpy the Hessian of $-\log f_{z_{\mathcal{X}}}$ evaluated at the point $c$, and may be seen as a surrogate of the Hessian of $-\log f_{\text{KME}}$ evaluated at the point $c$.

**Definition 2.** *Let $c \in \mathbb{R}^d$ such that $f_{z_{\mathcal{X}}}(c) > 0$. Define the matrix $\hat{H}$ by*

$$\hat{H} := -\Big(\text{Hess}(f_{z_{\mathcal{X}}}; c) f_{z_{\mathcal{X}}}(c) - \nabla_c f_{z_{\mathcal{X}}}^{\text{T}} \nabla_c f_{z_{\mathcal{X}}}\Big)/f_{z_{\mathcal{X}}}(c)^2, \tag{13}$$

*where $\text{Hess}(f_{z_{\mathcal{X}}}; c)$ is the Hessian matrix of the function $f_{z_{\mathcal{X}}}$ evaluated at $c$.*

Now, with $\Phi$ the random Fourier feature map with components $\phi_{\omega_j}$ defined by equation 2, the correlation function $f_{z_{\mathcal{X}}}$ from Definition 1 is

$$f_{z_{\mathcal{X}}}(c) = \frac{1}{\sqrt{m}} \mathfrak{Re} \sum_{j=1}^{m} \overline{z_j} \phi_{\omega_j}(c), \tag{14}$$

where $z_1, \ldots, z_m \in \mathbb{C}$ are the entries of the sketch $z_{\mathcal{X}}$. Thus

$$\nabla_c f_{z_{\mathcal{X}}} = \frac{1}{\sqrt{m}} \mathfrak{Re} \sum_{j=1}^{m} \overline{z_j} \omega_j \mathbf{i} \phi_{\omega_j}(c), \tag{15}$$

and the Hessian matrix $\text{Hess}(f_{z_{\mathcal{X}}}; c)$ is given by

$$\text{Hess}(f_{z_{\mathcal{X}}}; c) = -\frac{1}{\sqrt{m}} \mathfrak{Re} \sum_{j=1}^{m} \overline{z_j} \omega_j \omega_j^{\text{T}} \phi_{\omega_j}(c), \tag{16}$$

In other words, the matrix $\hat{H}$ defined by equation 13 can be numerically evaluated at a given $c$ using the values of the sketch $z_{\mathcal{X}}$ and the frequencies $\omega_1, \ldots, \omega_m$.

## C  Additional numerical experiments

### C.1  Growth of $L$ with the dimension $d$

In this section, we briefly illustrate that the number of initializations $L$ needed to achieve good performance suffers from the curse of dimensionality. For this, we conduct the same experiment across different dimensions $d \in \{2, 3, 5, 10, 20\}$: the dataset $\mathcal{X}$ is formed by $N = 10,000$ elements $x_i$ in $\mathbb{R}^d$, obtained through a generative model. The elements $x_i$ are i.i.d. samples from a mixture $0.5\mathcal{N}(c_1, \sigma_{\mathcal{X}}^2 \mathbb{I}_d) + 0.5\mathcal{N}(c_2, \sigma_{\mathcal{X}}^2 \mathbb{I}_d)$, where $c_1$ and $c_2$ are centers of the two normal distributions, $\sigma_0 > 0$, and $\mathbb{I}_d$ is the $d$-dimensional identity matrix. Figure 9 shows the RSE equation 10 of compressive clustering using the sketched mean shift approach with $m = 10d$ based on 10 realizations of the sketching operator for $L \in \{1, 10, 100, 1000\}$ and $d \in \{1, 2, 5, 10, 20\}$, for two different values of bandwidth $\sigma$. This empirically confirms that $L$ should grow with the dimension $d$.

### C.2  The case of CIFAR-10

In this section, we investigate whether the observations of Section 5.3 hold for the CIFAR-10 dataset. For this purpose, we perform experiments on the training set of the CIFAR-10 dataset, which consists of $N = 60000$

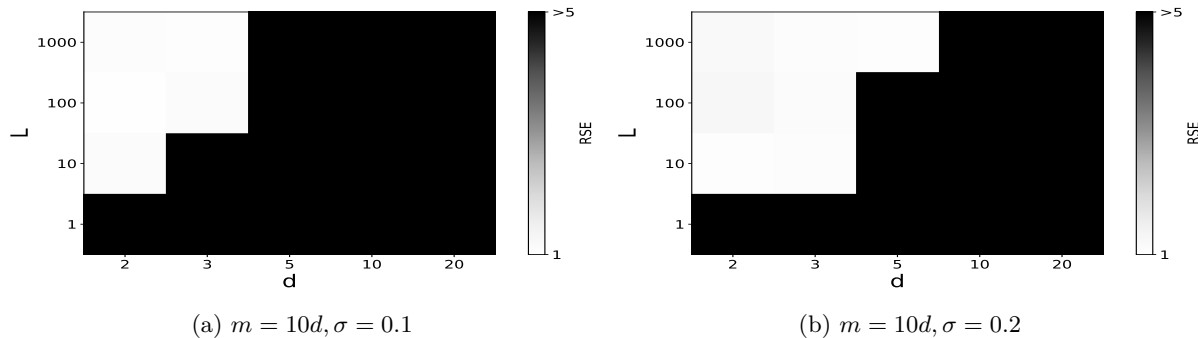

(a) $m = 10d, \sigma = 0.1$                      (b) $m = 10d, \sigma = 0.2$

Figure 9: The dependency between $L$ and $d$

images features with $k = 10$ classes. More precisely, we extracted features from the last fully connected layer (of dimension $D = 512$) of a trained ResNet18 (He et al., 2016). These extracted features are then reduced to a dimension of $d = 10$ using linear PCA. The network is trained on the training set of CIFAR-10 for 50 epochs with SGD with momentum 0.9, learning rate 0.1, learning rate decay 0.1, batch-size 512 and weight-decay $5 \times 10^{-4}$.

Figure 10 shows the RSE, defined by equation 10 based on Lloyd's algorithm of compressive clustering using CL-OMPR, and Algorithm 2 with mixtures of Diracs and the sketched mean shift approach with $L = 250$, based on 10 realizations of the sketching operator for various values of the sketch size $m$ and the bandwidth $\sigma$. Similarly to the experiments conducted in Section 5.3, we observe that Algorithm 2 outperforms CL-OMPR for $\sigma \leq 0.5$, and get a performance close to Lloyd's algorithm on a range of $\sigma$ that increases with $m$. We have observed running times that are of the same order of magnitude as in Section 5.3.

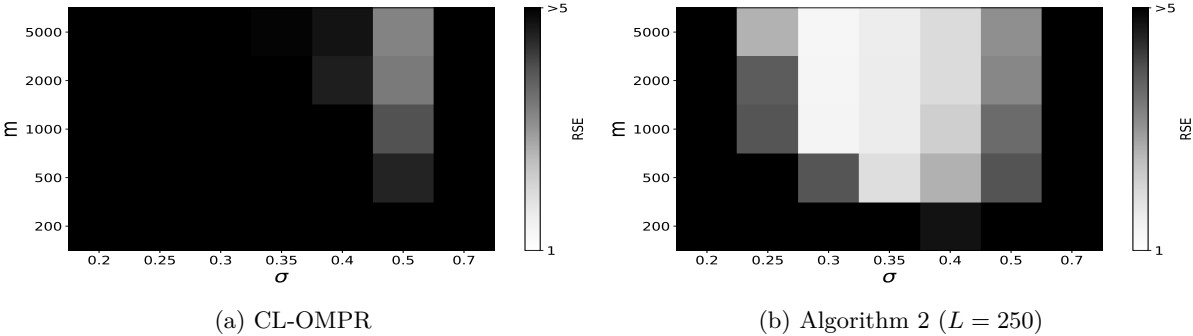

(a) CL-OMPR                      (b) Algorithm 2 ($L = 250$)

Figure 10: A comparison of CL-OMPR and Algorithm 2 on *the features of CIFAR-10*

