# OpenReview forum: "Sketch and shift: a robust decoder for compressive clustering"
_TMLR — Accepted by TMLR_

### Review · Reviewer_8iyN · 2023-12-21

**Summary Of Contributions:**

This paper studies compressive clustering, a task of clustering on sketches and decoding centroids from them. The focus is primarily on the standard heuristic CL-OMPR, revealing its limitations, particularly its failure to detect clearly separated clusters. To address this issue, the paper not only identifies the constraints of CL-OMPR but also introduces a novel algorithm with several improvements. Notably, two approaches are proposed to detect local maxima of the correlation function: a discretized approach and a sketched mean shift approach. The empirical evaluation, conducted on a synthetic dataset and the MNIST dataset, demonstrates the effectiveness of the sketched mean shift approach when compared to the original CL-OMPR.

**Audience:**

Yes

**Broader Impact Concerns:**

I do not have any concerns.

**Claims And Evidence:**

Yes

**Requested Changes:**

Please address the above listed weaknesses, that is, including a more diverse set of datasets, a more careful treatment of $L$, and reporting runtime.

**Minor issues:**
- P. 2, just after Eq.(4): "ths" -> "this"
- Sometimes 'Lloyd' is spelled as 'LLoyd,' whereas 'Lloyd' should be consistently used. Please correct this inconsistency.

**Strengths And Weaknesses:**

**Strengths**
- This paper is overall clearly written and easy to follow.
- Compressive clustering is an important research topic, and this paper explains its background and motivation well.
- This paper carefully analyzes the current approach (CL-OMPR) and proposes its improvement, which is a beneficial contribution to the community.

**Weaknesses**
- The empirical evaluation of this paper relies on only two datasets (one synthetic and one real-world). Given that the main contribution is tied to empirical results, it is necessary to include experiments conducted on a more diverse set of datasets.
- Furthermore, each dataset used is low-dimensional, with dimensions being 6 for the synthetic dataset and 10 for the real-world dataset. Since higher-dimensional datasets are more and more common in practical situations, conducting experiments on high-dimensional data is particularly desirable.
- While this paper explores the dependency of $L$, the limited diversity of datasets raises concerns about understanding how $L$ correlates with dataset properties. For example, if $L$ depends on the dimensionality and larger $L$ is needed for higher dimensional datasets, the effectiveness of the proposal is less convincing.
- The paper lacks the measurement of the actual runtime for each tested method. Including the runtime information would be valuable in assessing the efficiency of these methods.

---

> ### Author Response · Authors · 2024-02-12
>
> - Reporting runtime: you are right, although we did not record these during our experiments, the running time needed to compute a sketch from a dataset was negligible compared to the time needed to run CL-OMPR or Algorithm 2 on a sketch, which ranged between a few seconds and roughly ten minutes, depending on the sketch size m, the number of clusters k, and the value of parameter L. We will include precise figures in the final version after rerunning the experiments.
>
> - Higher dimensions : dimension 10 is pretty much the standard dimension of state of the art proofs of concept in sketched clustering. While our new algorithm is designed to overcome the important issues of CL-OMPR scrutinized in this paper, we do not expect this algorithm to be particularly good at overcoming issues related to a curse of dimension. We do have other ideas to handle this type of problem, which revolve around changing the feature map, e.g. with inspiration from the Nyström feature map of Chatalic et al. This is however clearly beyond the scope of the present contribution. Another algorithmic challenge is to handle large numbers K of clusters, as CL-OMPR is known to badly scale with K, as documented in the initial paper on Compressive K-Means of Keriven et al.
>
> - Treatment of L: we indeed observed L to increase rapidly with the dimension. Additional figures are in preparation to illustrate this in the appendix.
>
> - More diverse set of datasets: additional experiments on CIFAR10 are being run, to produce the analog of Figure 8.

---

### Review · Reviewer_hypU · 2024-01-12

**Summary Of Contributions:**

The paper addresses the problem of compressive learning. The goal is to reduce the whole dataset to a set of points (in the embedding space). A previous approach considered is CL-OMPR. It has a number of problems, including numerical instability and a requirement for a high-precision tuning of hyperparameters. Another alternative, LLoyd’s algorithm, is also computationally demanding. The authors propose improving the CL-OMPR approach and call the result Algorithm 2. We see that the proposed approach can recover the core set for artificial datasets and the classic MNIST.

**Audience:**

Yes

**Broader Impact Concerns:**

-

**Claims And Evidence:**

No

**Requested Changes:**

Inclusion of Analysis for Large Datasets:
Conduct a comprehensive analysis using larger datasets such as CIFAR10 or CIFAR100 to evaluate the scalability and performance of the proposed methods in more complex scenarios. This will provide a better understanding of how the model behaves with an increased number of classes and higher-dimensional data.

Proving Theoretical Properties:
Develop and present formal proofs for the theoretical properties discussed in the paper. A rigorous mathematical foundation will strengthen the validity of the partial discussions and support the empirical findings.

Connection to Core Set and Training Sample Transformation Literature:
Integrate references to the core set literature, explicitly drawing parallels and distinctions with the work presented in [1]. Furthermore, engage with the methodologies on transforming the training sample to a set of pseudoimages, as seen in [2, 3]. This should not just be a literature review but should also include a discussion of how these approaches relate to and can potentially enhance the current work.

Enhancement of Experiments:
Utilize the findings from [1], [2], and [3] to design additional experimental checks. Specifically, verify the efficacy of the new objects obtained from clustering, ensuring they contribute positively to the model's performance for a reduced sample. This validation will provide empirical evidence of the practical benefits derived from integrating insights from the referenced literature.

Diversification of Citations:
Broaden the range of references to include diverse research groups and papers. This not only addresses the issue of underrepresentation but also enriches the paper with a variety of perspectives and findings. It will demonstrate a more comprehensive engagement with the wider academic community and enhance the paper's credibility.

**Strengths And Weaknesses:**

Strength:
- an improvement over CL-OMPR is clear for considered datasets
- natural while interesting ideas on improvement at different steps

Weaknesses
- no analysis for large datasets (CIFAR10 or CIFAR100 would work here)
- no theoretical properties proved; only a partial discussion is conducted
- no connection to the core set literature [1] and the literature on transforming the training sample to a set of pseudoimages [2, 3]
- the findings from the papers above can be used to enhance experiments with additional checks that the new objects from clustering are beneficial
- the problem with the underrepresentation of other papers is highlighted by citing mostly papers from a single research group, more diversity would help


1. Guo, Chengcheng, Bo Zhao, and Yanbing Bai. "Deepcore: A comprehensive library for coreset selection in deep learning." International Conference on Database and Expert Systems Applications. Cham: Springer International Publishing, 2022.
2. Inoue, Hiroshi. "Data augmentation by pairing samples for images classification." arXiv preprint arXiv:1801.02929 (2018).
3. Naveed, Humza, et al. "Survey: Image mixing and deleting for data augmentation." Engineering Applications of Artificial Intelligence 131 (2024): 107791.

---

> ### Author Response · Authors · 2024-02-12
>
> - Large datasets: additional experiments on CIFAR10 are being run, to produce the analog of Figure 8. Regarding CIFAR100: just as CL-OMPR, our proposed algorithm is not expected to scale well with the dimension or the number K of clusters (see a discussion on this in the initial paper on Compressive K-means of Keriven et al), which would raise from K=10 to K=100 if switching to CIFAR100. We do have other ideas to handle these types of problem, which revolve around changing the feature map, e.g. with inspiration from the Nyström feature map of Chatalic et al. This is however clearly beyond the scope of the present contribution.
>
> - Theoretical properties: the existing literature on sketched clustering that we already cite has put a strong emphasis on establishing theoretical properties. These works essentially establish that the sketch “contains the relevant information to cluster”. There is also litterature proving that sketches can be made formally differentially private. Here our contribution is instead to document important practical limitations of CL-OMPR, which can fail to extract clustering information even when it is present in the sketch, and to propose relevant fixes. Our next priority is to address scalability with respect to dimension and number of clusters, which should be made easier thanks to the fact that the method is succesful with smaller sketch sizes. Establishing formal guarantees would of course come immediately next.
>
> - Coresets and Training Sample Transformation methods: relations and constrasts between coresets and sketched learning have indeed been documented in the sketched clustering / sketched learning literature, e.g. in the initial paper of Keriven et al on Compressive K-means. While coresets select a limited subset of samples aimed to be good representatives of the whole collection, sketching is somewhat more democratic and computes a unique sketch, depending equally on all samples, designed to capture the relevant information of the whole training collection to perform a given task. Sketching also allows to ensure differential privacy. To improve the self-containdness of our paper we added a paragraph about these aspects in the background section. We did not really understand in what sense the Training Sample Litterature [2,3] is related to our approach, but thank you very much for pointing out reference [1] on the DeepCore library. It will definitely be very useful in our next projects involving systematic comparisons with CoreSet approaches. Currently, conducting such extensive comparisons remains however far beyond the scope of our paper: our central goal is to address the main computational challenges of sketched clustering, bringing it from a nice (but limited) proof of concept much closer to a mature, turnkey machine learning tool.

---

### Review · Reviewer_XiFY · 2024-01-30

**Summary Of Contributions:**

The paper proposes a new heuristic algorithm for compressive clustering of data. Through carefully designed synthetic experiments, the authors are able to identify the deficiencies of the baseline method (CL-OMPR) and propose and algorithmic fixes in several dimensions. The modified algorithm is then further tested on synthetic data and MNIST dataset.

**Audience:**

Yes

**Claims And Evidence:**

Yes

**Requested Changes:**

For requested changes, see "Weaknesses" section above.


**Typos:**

1. Page 2 "In Section 3, we show conduct", "ths non-convex", Page 6 "litterature", Page 8 "we we perform",

2. “In this section, we investigate whether the observations of Section 5.3 hold for real datasets” Should be Section 5.2?

**Strengths And Weaknesses:**

**Strengths:**

1. The paper is very instructive, demonstrating the issues of the existing approach in several aspects (e.g., vanishing gradients, multiple local maxima, lack of exploration). Each of these aspects is then addressed by certain modification: normalization of gradient ascent step by the function value, replacing the mixture of Diracs with mixture of Gaussians.

2. The performance gains presented in Section 5 (especially Figure 5) are substantial.

3. The experiments are mostly performed in a controllable environment with small datasets, which allows to understand the issue and potential gains of the new algorithm. The error bars are included where appropriate.

 **Weaknesses:**

1. The illustration/description of experiments in section 5.2 is unclear to me. Figure 7 has several duplicate captions for different plots, which makes it difficult to understand the results. I assume the left plots (a), (c) and (e) correspond to a separable case and the right plots are for non-separable?

2. In section 5, the paper suddenly introduces RSE metric instead of previously used MSE metric. Is this a standard metric in the field? It's unclear to me why Figure 5 uses MSE and the rest of experiments use RSE. Some explanation/motivation would be helpful.

3. It would be useful to include the running time and memory cost of the proposed algorithms.

**Questions:**

1. What is exactly meant by “phantom local maxima”? Cannot find the definition for this term in the paper.

2. What is the connection between CL-OMPR and Frank-Wolfe method (stated in the abstract)? Does the connection also hold for Alg. 2?

---

> ### Author Response · Authors · 2024-02-12
>
> - Clarity of Figure 7: the current layout is indeed unclear, thank you for pointing this out. We updated it in the revised version, hopefully this is now much clearer.
>
> - RSE vs MSE: the RSE is just a “normalized” MSE, relative to a reference. It is commonly used in the literature on sketched clustering. In Figure 1(a) we displayed the MSE because its value is informative in comparison to the intra-cluster variance and inter-cluster squared distance of the data displayed on Figure 1(b). We added a comment about this in the caption of Figure 1. We realize thanks to your comment that it is better to use the RSE starting from Figure 5, and we therefore also anticipated the location of its definition.
>
> - Running time and memory: you are right, although we did not record these during our experiments, the running time needed to compute a sketch from a dataset was negligible compared to the time needed to run CL-OMPR or Algorithm 2 on a sketch, which ranged between a few seconds and roughly ten minutes, depending on the sketch size m, the number of clusters k, and the value of parameter L. We will include precise figures in the final version after rerunning the experiments.
>
> - “phantom local maxima”: we meant “spurious local maxima” and the revised version switched to this standard terminology.
>
> - CL-OMPR vs Franke-Wolfe: thanks for pointing out that we mentioned this connexion in passing in the abstract without coming back to it in the main text. This is now fixed with some comments below Equation (4).

---

### Author Response · Authors · 2024-02-14

Our computational server is currently under maintenance, the additional experiments will take longer than anticipated.

---

> ### Comment · Action_Editor_sUmK · 2024-02-19
> **Followup questions**
>
> Dear reviewers,
>
> While the authors work on these experiments, do you (the reviewers) have any responses to the authors' response? One neat thing about the OpenReview framework is that you can have a back-and-forth conversation, which is part of how we can cut the review time down.
>
> In particular, the **official reviews are due in 8 days from now**.  If you anticipate having any significantly negative aspect of the review, please bring it up now as part of a conversation.
>
> Thanks
>
> Stephen (AE)

---

> > ### Comment · Action_Editor_sUmK · 2024-02-19
> > **Criteria**
> >
> > Now is also a good time to remind all of us about the acceptance criteria.
> >
> > "The acceptance decision for a submission is based on the answers to the following questions:
> > - Are the claims made in the submission supported by accurate, convincing and clear evidence?
> > - Would at least some individuals in TMLR's audience be interested in knowing the findings of this paper?
> >
> > "Papers should be accepted if they meet the criteria, **even if the contribution or significance of the work is modest**.
> >
> > "Papers that should **not** be accepted include papers that **make bold statements unsupported** by empirical or rigorous evidence, papers that aren’t clearly written, papers that **incorrectly claim novelty** over existing published work, and papers that merely re-implement an idea that has already been reproduced before."
> >
> > and from https://jmlr.org/tmlr/acceptance-criteria.html,
> >
> > "[Criterion]: Would some individuals in TMLR's audience be interested in the findings of this paper?... Crucially, [this criterion] should not be used as a reason to reject work that isn't considered “significant” or “impactful” because it isn't achieving a new state-of-the-art on some benchmark."

---

### Author Response · Authors · 2024-02-29
**Revision**

Dear reviewers,

We have added two additional numerical simulations in the appendix. The first one studies the dependency between the number of initializations $L$ and the dimension $d$, while the second studies the case of the CIFAR-10 dataset. Moreover, we have included the order of magnitude of the running time of our algorithms.

---

### Decision · Action_Editor_sUmK · 2024-04-01

**Recommendation:** Accept as is

**Comment:**

The reviewers mostly agreed that this was a clear improvement over the baseline method, CL-OMPR, and one reviewer found the paper very instructive in diagnosing the problems with the baseline method and deriving the new algorithm.  There were some requested changes by reviewers, and in the AE's opinion, the authors suitably addressed these requests, including running more experiments and adding more text about alternative approaches (such as coresets).  No reviewer raised issues about novelty nor correctness. The biggest criticisms were that (1) this is a *narrow* paper, designed only for a very specific setup of compressive clustering, and (2) that the method is limited to moderate dimensions.  In response, I think that (1) there is still a sufficient TMLR audience for this paper, and given the novelty, it meets the TMLR criteria, and (2) the dimension limitation is a drawback of this family of methods but there is still a clear use-case for these kinds of methods (as well as room for future improvement).

**Audience:**

Compressive learning does not have a huge audience itself, but it aims to solve problems that are common in data science. Overall, it's a topical subject.  I think there is clearly enough of an audience to find this paper useful.

**Claims And Evidence:**

The paper addresses compressive learning, and in particular compressive clustering. The "compression" or "sketch" in this framework (as introduced in 2017) is a *democratic* (unweighted) sum of sketches of the data points, in contrast to sampling methods like coresets. Within this compressive framework, CL-OMPR is the most popular clustering method. This paper (1) shows that CL-OMPR has limitations, and (2) it proposes some fixes to these limitations. To justify these claims, the paper uses mathematical arguments and numerical experiments, and connections to other fields. It does not focus on a big single theorem, but it still uses mathematical arguments.  None of the reviewers raised issues with the evidence, other than requesting more numerical simulations (which were done).